# Toxicity of Metal Ions Released from a Fixed Orthodontic Appliance to Gastrointestinal Tract Cell Lines

**DOI:** 10.3390/ijms24129940

**Published:** 2023-06-09

**Authors:** Ksenija Durgo, Sunčana Orešić, Marijana Rinčić Mlinarić, Željka Fiket, Gordana Čanadi Jurešić

**Affiliations:** 1Department of Biochemical Engineering, Faculty of Food Technology and Biotechnology, Universtiy of Zagreb, Pierrotijeva 6, 10000 Zagreb, Croatia; kdurgo@pbf.hr (K.D.); suncanaoresic@gmail.com (S.O.); 2Private Orthodontic Practice, Katarine Zrinske 1b, 23000 Zadar, Croatia; orto.rincic@gmail.com; 3Division for Marine and Environmental Research, Ruđer Bošković Institute, Bijenička 54, 10000 Zagreb, Croatia; 4Department of Medical Chemistry, Biochemistry and Clinical Chemistry, Faculty of Medicine, Universtiy of Rijeka, B. Branchetta 20, 51000 Rijeka, Croatia

**Keywords:** metals, orthodontic appliance, prooxidant effect, cytotoxicity, genotoxicity, gastrointestinal tract cell lines

## Abstract

The mechanism of toxicity and cellular response to metal ions present in the environment is still a very current area of research. In this work, which is a continuation of the study of the toxicity of metal ions released by fixed orthodontic appliances, eluates of archwires, brackets, ligatures, and bands are used to test the prooxidant effect, cytotoxicity, and genotoxicity on cell lines of the gastrointestinal tract. Eluates obtained after three immersion periods (3, 7, and 14 days) and with known amounts and types of metal ions were used. Four cell lines—CAL 27 (human tongue), Hep-G2 (liver), AGS (stomach) and CaCo-2 (colon)—were treated with each type of eluate at four concentrations (0.1×, 0.5×, 1.0×, and 2.0×) for 24 h. Most eluates had toxic effects on CAL 27 cells over the entire concentration range regardless of exposure time, while CaCo-2 proved to be the most resistant. In AGS and Hep-G2 cells, all samples tested induced free radical formation, with the highest concentration (2×) causing a decrease in free radicals formed compared to the lowest concentrations. Eluates containing Cr, Mn, and Al showed a slight pro-oxidant effect on DNA (on plasmid φX-174 RF I) and slight genotoxicity (comet assay), but these effects are not so great that the human body could not “resist” them. Statistical analysis of data on chemical composition, cytotoxicity, ROS, genotoxicity, and prooxidative DNA damage shows the influence of metal ions present in some eluates on the toxicity obtained. Fe and Ni are responsible for the production of ROS, while Mn and Cr have a great influence on hydroxyl radicals, which cause single-strand breaks in supercoiled plasmid DNA in addition to the production of ROS. On the other hand, Fe, Cr, Mn, and Al are responsible for the cytotoxic effect of the studied eluates. The obtained results confirm that this type of research is useful and brings us closer to more accurate in vivo conditions.

## 1. Introduction

Although there are still numerous studies and tests on the biocompatibility (corrosion behavior and toxicity) of orthodontic materials, the results obtained are not uniform and no clear conclusions can be drawn because of the increasing variability of the materials, their composition, and the manufacturing processes [1,2]. Therefore, it is still useful to test each type of orthodontic material/alloy individually, make different combinations of devices from different materials, immerse and elute them in different media, and test the toxicity of the different/specific combinations by using in vitro models.

Main base metal orthodontic alloys commonly used for orthodontic appliances are stainless steel (SS), cobalt−chromium, nickel−titanium (NiTi), and beta titanium [3].

Fixed orthodontic appliances, the most common and effective form of orthodontic treatment, consist of archwires, bands, ligatures, and brackets that are attached to the teeth with a special adhesive, usually acrylic resin [4]. In this study, we decided to combine NiTi archwires and SS brackets, bands and ligatures as components of a fixed orthodontic appliance and use them in the cytotoxicity study. NiTi alloy is used in orthodontics mainly for wires. Its unique properties—superelasticity and shape memory effect—are limited to a narrow composition range close to its equiatomic composition. All stainless steel parts are made of austenitic steel, which is graded differently according to specific requirements of every component and has good corrosion resistance under many different environmental conditions [3].

The oral cavity is a very dynamic environment, subject to constant changes in composition, temperature, pH, and saliva quantity [3,4]. Changes in pH, as we see in patients with poor oral hygiene and plaque accumulation, can affect the stability of the used materials by accelerating corrosion processes [5,6]. Glossitis, metallic taste in the mouth, dry lips, inflammatory redness, and hypertrophy of the gums are commonly observed in patients and are the consequences of the release of these metals from dental alloys into saliva. Even if the device itself does not cause serious damage to the oral cavity, it may make it more susceptible to damage and toxic substances later in life [1,7,8]. Although studies on the cytotoxicity of these metals and orthodontic appliances have been conducted frequently and most show that cytotoxicity is negligible, it cannot be ruled out that low concentrations of metals are sufficient to cause biological changes in the oral mucosa, especially because they are in the form of mixtures [1,9,10]. The mucosal tissue of the oral cavity can absorb these low levels of metal ions over a long period of time, leading to changes in the genome of the cells of the oral cavity. In addition, metals are not biodegradable and can have irreversible toxic effects due to their accumulation in tissues. In addition, cobalt and nickel have been shown to be carcinogenic to mucous membranes [1]. These metals can accumulate locally and be found in body fluids and tissues unrelated to the oral cavity. In general, exposure to metal ions causes various pathological effects such as inflammatory responses, alterations in the oxidative stress response, increased lipid peroxidation, and changes in DNA repair mechanisms [1].

Nowadays, fibroblast cell lines (periodontal ligament, gingival, or dermal [9,10,11,12,13,14]), gastrointestinal tract cell lines [15,16], and sometimes yeast [17,18] are most commonly used for testing. Research has focused mainly on NiTi or stainless (arch)wires [2,13,19,20], brackets [2,12,14], and bands [2,9,10,16,21].

The aim of this study is to determine the cytotoxic, prooxidant, and genotoxic effects of metal ion eluates prepared separately from each part of a fixed orthodontic appliance (archwires, brackets, ligatures, and bands). The eluates were prepared in artificial saliva after three different exposure times (3, 7, and 14 days) and determined qualitatively and quantitatively by ICP-MS. Thus, the metal ion composition was known for each experimental eluate used. Cell lines from the oral cavity, stomach, liver, and colon, selected to mimic the passage of saliva through the gastrointestinal tract, were used as the biological test system. Possible correlations between ion release profiles and observed prooxidant activity, cytotoxicity, and genotoxicity were investigated. The experimental data are used to determine whether the selected fixed orthodontic appliance is safe to use and what side effects it may cause.

## 2. Results

This work is a continuation of the study on the toxicity of metal ions released by fixed orthodontic appliances, and for a better understanding it is necessary to mention some of the main conclusions of the previous study [22]. It was found that metal ions were released from all parts of dental alloys studied. The results from ICP-MS indicate that the bands released the highest amounts of metal ions in artificial saliva samples. Bands immersed in artificial saliva for 7 days (B7D) released more than 6 mg/L, while bands immersed for 3 (B3D) or 14 days (B14D) released 3.7 and 3 mg/L, respectively, of all ions detected. According to the manufacturer’s specifications, the band should contain 17–19% Cr, 8–10.5% Ni, and the rest Fe. Mn, although not specified, was also detected in all the samples tested. Although Ni accounts for only about 10% of the bands, the content of this ion in artificial saliva samples is very high—highest in sample B14D— accounting for one-third of all detected ions. In sample B7D, which has the highest content of all detected ions, iron accounts for one-third of all detected ions, but Fe and Ni together account for more than 3.5 mg/L.

The content of released nickel ions in the samples of the archwires is slightly higher than the value published by Rinčić-Mlinarić et al. [15] under similar experimental conditions after an exposure time of 14 days (38.61 μg/L in this work versus 15.1–30.4 μg/L in the cited work). In archwires eluates, the content of Ni was highest after 7 days of elution (271 μg/L approx., is consistent with the results of Kuhta et al. [23] and Hwang et al. [24]), while the content of all certain metals was highest after 14 days of elution (1261 μg/L, approx.). According to the results presented in Petković Didović et al. [22], immersion of NiTi archwires in artificial saliva did not show pronounced morphological changes (SEM and EDX). Highly polished NiTi wires did not appear to contain an oxide layer in the as-received condition, but the gradual decrease in eluted Ti concentrations indicated that a thin protective oxide layer (composed mainly of TiO_2_) developed during the time of immersion. The immersion in artificial saliva did result in the development of an adherent layer on the surface of the SS bands and ligatures, but based on the obtained results was not of the same type. The quantity of released ions correlates with the formation of layer.

The bands had the highest risk in terms of release of (heavy) metals. During the 7-day elution period, significant amounts of chromium, iron, and nickel, together with aluminum and lead, and copper, zinc, and cobalt were released in saliva.

### 2.1. Determination of Cytotoxicity

Increased concentrations of metal mixtures may pose a serious threat to cellular stability and function of cellular macromolecules. In this work, the potential cytotoxic effect of eluted saliva samples was investigated using the neutral red method, which measures the ability of viable cells to take up the supravital dye neutral red and bind it in lysosomes. Tongue cells were the most sensitive cells; all saliva samples (eluates) tested were toxic to this cell line at all concentration ranges tested. In addition, a most intriguing and interesting response of CAL27 cells to metal ion treatment was noted. For most samples, viability was lowest at the lowest dose tested (0.1×), with a tendency to increase at a dose of 2×. One of the possible explanations for this phenomenon is that Fe depletion leads to G(1)/S arrest and apoptosis, so the presence of increased iron may stimulate cell proliferation [25]. Alberto et al. [26] have shown that manganese can also stimulate cell proliferation. Thus, it is possible that these metals, once present in the mixture, have effects on cell division processes and stimulate cell proliferation at very low concentrations (the dose was 0.1×). Considering the cytotoxic effect of the eluates on other cell lines, only Hep 27 cells and a decrease in viability after treatment with eluates of B14D stand out. No cytotoxic effect on stomach and colon cell lines was detected (Figure 1).

### 2.2. Determination of Free Radicals

The presence of free radicals tested in four gastrointestinal track cancer cell lines after exposure to eluates of metal ions released from different parts of fixed orthodontic appliance (archwires, brackets, ligatures, bands, and all these parts combined) at four different concentrations (0.1×; 0.5×, 1.0×, and 2.0×) during the exposure period (3, 7, and 14 days) is presented in Figure 2.

There is a visible linear or arch-shaped trend of decreased presence of free radicals in the AGS, HEP-G2, and CAL27 cell lines, as the eluate tested concentrations increased. Most often in low tested concentration the effect is prooxidative, while in the 2× concentration the effect is antioxidative. Surprisingly, the highest ROS production was noticed in AGS A7D, 0.5× concentration.

### 2.3. Testing for DNA Damage

In this work, the plasmid DNA cleavage assay using supercoiled φX-174 RF I plasmid DNA was used to determine whether saliva increases oxidative DNA damage after exposure to UV light. UV production of oxygen radicals leads to oxidative damage and single and double strand breaks.

Figure 3 shows the results of the influence of the studied saliva samples on DNA stability. All samples showed a slight prooxidant effect on DNA (lower ratio of supercoiled/linear form of the plasmid compared to the negative control, but higher ratio compared to the positive control).

### 2.4. Determination of Genotoxicity

Not all samples are eligible for the genotoxicity test of the comet assay, but only those that exhibit some prooxidant activity but also result in cell survival greater than 80%. Based on these parameters, specific samples with specific concentrations were selected and tested only on the appropriate cell lines (Figure 4). Ligatures 14 days in concentration 1.0× on the CAL 27 cells; then, bands 3 and 7 days, in all samples 0.5× concentration, on the AGS; and finally bands 7 days in 0.5× concentration, on the CaCo2 cell line showed genotoxic effect (having greater values than control).

Figure 4 shows the results of the genotoxic effect.

These samples have increased concentration of iron, nickel, and chromium compared to the control: in L14D, a certain amount of eluted Al contributes to genotoxicity [27]; in B7D, Fe (>2.3 mg/L) indicates ferroptosis [28]. Synergistic effects of metals can also lead to increased genotoxicity of the samples [29].

Although a significant change compared with the control sample is noticed, none of the samples tested on Hep-G2 cells caused an increase in tail intensity, but instead a decrease or diminished value. In all these experiments, tail intensity was very high in the control samples. The most sensitive cells were CAL 27, on which most eluates had a toxic effect throughout the concentration range, regardless of exposure time. The consequences of the cytotoxic effect were also evident in the prooxidation effect, which could not be detected in these cells due to the increased toxicity.

### 2.5. Principal Component Analysis of the Potential Correlation

Principal component analysis (PCA) was used to determine the potential correlation between the parts of orthodontic appliance (archwires, brackets, ligatures, and bands in three time periods were used as variables) and the analyses performed (cytotoxicity and the presence of free radicals in four different cell lines, the four most highly expressed metals and aluminum eluted from the orthodontic appliances, and the DNA damage caused by eluted metals at four different concentrations were used as cases). Figure 5 shows the values of the principal components and their contribution to the total variance.

The number of components used in the PCA was determined using the Cattell scree test. On this basis, three principal components (expressed with eigenvalue > 1) were used for further analysis, and their values along with their contribution to the total variance are shown in Table 1. The first component explains 53.36% of the total variance, the second 26.42%, and the third 9.57%.

The eigenvectors of the correlation matrix (presented in Table 2) were used to interpret the principal component analysis. Component 1 is defined mainly by the brackets 3D and 14D and the ligatures 3D, 7D, and 14D. Component 2 is defined by the archwires 14D and the bands 3D, 7D, and 14D. Component 3 is defined by only three variables: archwires 3D and 7D and brackets 7D. It is worth noting that all variables defining component 1 are on the left side of the factor plane and have a negative value. Combining the results of both factor planes shows that the variables defining component 2 are related to the iron and nickel content of these samples, while that of the cases correlate genotoxicity with the content of manganese, chromium, and aluminum.

## 3. Discussion

By definition, dental alloys are a mixture of two or more metals with more or less affinity for migration in the environment. The strength and concentration of migration of the metals depends largely on the composition of the dental alloy and the physicochemical properties of the environment. The grain structure of the alloy also predisposes to corrosion and often plays a different role in different dental alloys [2,19,30]. Since there is no “ideal” dental alloy from which no metals migrate, the question is which metals migrate and what is the migration rate from the alloys to the environment? Finally, the most important question is how the constant contact of the buccal cells and the epithelial cells lining the digestive system with the metal mixture affects the stability of the cellular macromolecules.

The tendency of a metal to corrode is defined by its electrode potential, so metals with higher negative potential (such as nickel, iron, chromium, manganese) are more likely to corrode. Noble metals with positive potential, such as gold or platinum, are less reactive. The most negative potential of abovementioned metals is chromium, but it is often added into dental alloys since it reacts in the way it forms oxide that will, in further reactions, protect dental alloy, and prevent flow of metals from alloys [30]. The explanation for the unpredictable release of heavy metals from the dental alloys and bands obtained in this study, which does not depend on the days of incubation in saliva, lies in the mechanism mentioned above, which includes the manufacturing process and the type of alloy, the progress of galvanic corrosion, and the formation of a passive protective or adherent layer over time [22].

Cytotoxicity is only preliminary information about the potential for a chemical to disrupt the normal functioning of the cell and, ultimately, to cause cell death, so it is very important to determine the possible mechanisms responsible for this effect. The results of in vitro analyses may be indicative of the effects observed in vivo, although the presence of a cytotoxic effect in vitro does not mean that the material is toxic for use in vivo. This is due to the conditions of laboratory testing, which differ significantly from the complex oral environment [1,16]. Nevertheless, the absence of a cytotoxic effect is a guarantee of a good clinical response [3]. The SS bands released significant amounts of nickel, iron, and chromium. According to the data from several studies [22,31,32], stainless steel is less desirable as a biomaterial for the production of fixed orthodontic appliances because it releases the largest amount of nickel and chromium and is the least biocompatible material. The total amount of metals released from archwires, brackets, and ligatures was significantly lower than for bands, in this study. Similar findings were obtained by Wendl et al. [2]. The highest concentration of nickel measured was 1497 µg/mL (B7D incubation), iron 2385 µg/mL (B7D), chromium 723 µg/mL (B7D), and manganese 177 µg/mL (B14D). Toxicity data for the most abundant metals suggest that these concentrations are below those that cause toxic effects [29]. Wen et al. [33] tested various concentrations of nickel on seven cell lines and showed that concentrations below 1200 mg/mL were not toxic. None of the samples tested in this work exceeded this concentration, so it can be concluded that nickel is not responsible for the toxic effects of saliva. Costa et al. [32] found that the highest nickel concentration was detected in the extracts of an AISI SS bracket after an exposure time of 42 days (4.46 ± 0.68 g/mL) and in the low-nickel extracts of the SS bracket after 63 days (0.07 ± 0.01 g/mL). Maximum manganese release was also observed from the AISI 304 SS bracket extracts after 42 days (0.90 ± 0.05 g/mL) and from the low-nickel SS bracket extracts after 21 days (0.11 ± 0.02 g/mL), indicating that metal elution is highly dependent on the material used. The researchers found varying degrees of cytotoxicity of SS and low-nickel SS bracket extracts in L929 cell cultures. Neither the extracts from the SS bracket nor from the low-nickel SS bracket significantly altered cell viability, but a slight decrease in metabolic activity was observed with the 42-day SS bracket extract. Extracts from the low-nickel SS bracket did not result in significant changes in cell metabolism, and the metabolic activity of cells was not altered by any of the 21-day extracts [32]. Kanaji et al. [34] demonstrated that chromium has no toxic effect on cells at concentrations less than 0.5 mM, equivalent to 25.998 µg/mL, but metal ions released from metal-on-metal bearing surfaces have potentially cytotoxic effects on MLO-Y4 osteocytes in vitro. Wu et al. [35] showed that exposure to 10 µM potassium dichromate (K_2_Cr_2_O_7_) significantly decreased the viability of HK-2 cells after 24 and 48 h of incubation and induced the intracellular formation of ROS. They also demonstrated that the expression levels of markers that activate the apoptotic pathway, including cleaved caspase-3 and poly(ADP-ribose) polymerase, were significantly upregulated after K_2_Cr_2_O_7_ treatment of HK-2 cells. The threshold for acute oral toxicity of chromium (III) was 1900–3300 mg/kg [35]. The saliva obtained from the elution of the bands after 7 days with the highest chromium concentration was toxic to the tongue cells, so one of the reasons is exceeding the 700 µg/mL concentration. Concerning toxicity, it should be kept in mind that an important point is the synergistic and additive effect in toxicity when two or more metals are present in the same mixture. Rinčić Mlinarić et al. [15] demonstrated that Ni and Ti ions alone do not have a major cytotoxic effect, but their combination does, indicating their synergistic effect. Although for most of the samples studied in this work the content of individual metals (with the exception of the bands) did not exceed the toxic concentration, they proved to be toxic to the epithelial cells of the tongue. One of the explanations for this is the synergistic effect of the different ions present in these samples. On the other side, considering that the CAL cells are the cancer cells whose viability decreases (from 60% to 80%) when treated with released metal ions, the question arises whether released ions from orthodontic appliances can be considered as anticarcinogenic, which puts the effect of released metal ions into a completely different light.

Nickel, iron, and chromium, the most common metals found in saliva after 3, 7, and 14 days of elution, catalyze free radical formation through a Fenton-like reaction, whereas cadmium and other redox-inactive metals can trigger the formation of ROS through an indirect mechanism. ROS include both free radicals and non-free radical oxygen-containing molecules, such as hydrogen peroxide (H_2_O_2_), superoxide (O_2_^−^), singlet oxygen (1/2O_2_), and the hydroxyl radical (HO). On the other hand, free metal ions can enhance oxidative stress because they can amplify the redox cycle while the metal ion can accept or donate a single electron. This process catalyzes reactions in which reactive radicals can be generated and reactive oxygen species can be formed [29]. In this work, all the tested samples induced the formation of ROS (Figure 2). In some cases, the highest concentration (2×) resulted in a decrease in free radicals compared to the control. In the case of epithelial cells of the tongue, one of the reasons could be the cytotoxic effect of the tested samples. In the case of hepatocarcinoma cells and gastric cell lines, the reason could be that when the cells are exposed to a certain concentration of a metal mixture, they begin to develop an antioxidant system that degrades the already-formed reactive species. It is known that the endogenous defense system includes enzymatic and non-enzymatic antioxidants such as superoxide dismutase, glutathione peroxidase, catalase, peroxiredoxins, glutathione (GSH), thioredoxin, uric acid, and a system for repairing oxidative damage to molecules [29]. Thus, different mechanisms are triggered at lower and higher concentrations of released metal ions.

The most resistant cell line was the adenocarcinoma cell line of the colon. CaCo-2 cells have been an excellent model system for many different tests—as a monolayer, it mimics the human intestinal epithelium. Although CaCo-2 cells express many different proteins (for transport and efflux) and enzymes (phase II conjugation) to model a variety of transcellular pathways and metabolic transformation of test compounds, they lack expression of cytochrome P450 isozymes (particularly CYP3A4) [36].

The samples containing a high concentration of nickel, but also cobalt, zinc, and chromium, exhibited the highest prooxidant activity, which increased with the duration of incubation in saliva, as did the metal concentration. In addition, all samples that had significantly higher concentrations of chromium and nickel compared to the control, especially those with exposure times of 7 and 14 days, showed the potential to cause increased ROS production. Oxidative DNA damage is one of the most common consequences of exposure to external environmental factors or endogenous genotoxic substances. These reactions are associated with the action of ROS such as hydroxyl radicals, superoxides, peroxides, or simple oxygen. In the combined UV/H_2_O_2_ process, UV radiation activates H_2_O_2_, leading to the formation of the OH radicals [37,38]. The effectiveness of the UV/H_2_O_2_ process depends on several conditions that affect its ability to degrade organic molecules. The conversion of the supercoiled form of plasmid DNA to an open-circular form is used as an indicator of DNA damage [38]. The structure of supercoiled DNA is a good model for studying DNA damage because it is very sensitive to environmental conditions, because even a single-strand break (which occurs when the plasmid is exposed to a hydroxyl radical) in a supercoiled DNA causes the super helical structure to loosen and the plasmid to assume a loose form [39,40]. Double-strand breaks open the plasmid and convert its structure to a linear form. When there are many double-strand breaks, the plasmid is fragmented into linear duplexes that are shorter compared to the original length [37,38]. All these changes can be analyzed and quantified by gel electrophoresis and gel analysis software (Gel Analyzer 19.1). Figure 3 shows the results of the influence of the studied saliva samples on DNA stability; all samples showed a slight prooxidant effect on DNA (lower ratio of supercoiled/linear form of the plasmid compared to the negative control, but higher ratio compared to the positive control). Concerning the metals present in saliva, the results obtained from this work, indicating a slight prooxidative effect of almost all tested eluates, were expected since previous research has also confirmed the role of the prooxidant activity of metals in the development of DNA damage, using the M13 mutation assay and DNA sequencing methods and the plasmid linearization method [41,42]. Thorough studies on DNA damage mediated by some first-row transition metals (iron, cobalt, nickel, copper, zinc, chromium, manganese) under oxidative stress conditions are presented in the work of Angelé-Martínez et al. [42]. Higher concentrations of cobalt, chromium, and nickel were also found in the appliances used in this study, which could be related to the prooxidant effect obtained. In addition, the plasmid–aluminum relationship is also confirmed.

The metals that enter saliva during the initial period after orthodontic appliance insertion can cause DNA damage, but to a relatively small extent because the proportions of the supercoiled plasmid form are still closer to the negative control than to the positive control. Genotoxicity is often associated with oxidative stress and free radical formation [41]. Loyola-Rodriguez et al. [14] demonstrated that all types of orthodontic brackets, regardless of their components, cause DNA damage when their eluates are exposed to human gingival fibroblasts using the comet assay. The results of the comet test performed in this work, expressed as tail length, are shown in Figure 4. Only samples showing some prooxidant activity and having a cell survival rate higher than 80% are eligible for the genotoxicity test of the comet assay. Genotoxicity was mainly found in the samples with increased concentration of iron, nickel, and chromium compared to the control. The eluate incubated with all parts for 3 days has nine times more aluminum than the control, which also contributes to genotoxicity [27]. Synergistic effects of metals can also lead to increased genotoxicity of samples [29,31]. Samples with bands have up to 100 times higher concentration of nickel and chromium, so significant genotoxicity of these samples is expected [41,42], but the results did not show significant genotoxicity in all samples. It is possible that the DNA repair mechanism was activated in the cells, so the effects of the metal on DNA are not so visible [43]. The samples obtained after eluting the bands and all parts combined for 3 and 7 days, respectively, showed an increase in genotoxicity levels. Samples obtained after 14 days of elution showed stagnation or a slight decrease in tail intensity (toxicity is detected only for CAL27 cells). This is consistent with the results that Petković Didović et al. [22] obtained since the concentration of certain metals such as nickel and chromium, which have a genotoxic effect, decreases on the 14th day compared to the 7th day of elution. This result is in accordance with the previously conducted studies, which, as mentioned above, also showed that genotoxicity increases with the duration of metal exposure, but also that it decreases again after long-term elution of dental alloys [43,44,45].

Unusual results in Hep-G2 cells (very high value of tail intensity in control samples and diminished values in treated samples), was noticed, probably due to activation of DNA repair mechanisms in these cells. Liver cells rich in proteins that have affinity for binding various metals are able to reduce the concentration of metals in free form that could have a genotoxic effect [43,44,45]. These results could be encouraging because the genotoxicity of metals from orthodontic appliances is not so great that the human body cannot “defend” itself against them.

From the results of the principal component analysis (PCA), it can be concluded that Fe and Ni are responsible for the production of ROS, while Mn and Cr have a major impact on hydroxyl radicals causing single-strand breaks in supercoiled plasmid DNA, in addition to the production of ROS. On the other hand, Fe, Cr, Mn, and Al are responsible for the cytotoxic effect of the eluates studied. There is a significant correspondence between the generation of ROS in the cells and the single strand breaks in the plasmid. Cr, Mn, and Al are highly responsible for genotoxicity as these cause linearization of the plasmid at all concentrations tested. This result is in agreement with the data found in the literature on the toxicity of Cr (VI), which is mainly present in stainless products [31,41,46].

The only limitation in the study that the authors were able to identify is the fact that the experiments were performed on cell lines that are different from normal cells. However, a negative control was included in all experiments, and all results obtained are presented in comparison with the control. All observed changes are the result of exposure to the eluates and the different metal concentrations in the eluates. Finally, the use of cell lines in toxicological studies is widely accepted as a biological test system, and the results can be well interpreted by appropriate statistical analysis.

## 4. Materials and Methods

### 4.1. Materials

#### 4.1.1. Cell Lines

Epithelial lingual (CAL 27), hepatic (HepG2), colon (CaCo2), and stomach carcinoma (AGS) cell lines were provided by the European Collection of Authenticated Cell Cultures (ECACC). The cell lines were used to determine the cytotoxic, prooxidant, and genotoxic effects of dental alloy eluates. Cells were grown as monolayer cultures in growth media recommended by the cell line supplier and supplemented with 10% fetal bovine serum (GIBCO, New York, NY, USA), 4500 mg/L glucose, and 1% penicillin/streptomycin. The supercoiled plasmid φX-174 RF I (Promega, Madison, WI, USA) was used to determine the prooxidative effect of the dental alloy eluates on DNA.

#### 4.1.2. Orthodontic Appliances

Orthodontic archwires used in the study were nickel and titanium alloy (rematitan^®^ LITE ideal arches, φ 0.43 × 0.64 mm/17 × 25, Dentaurum, Baden-Wurttemberg, Germany), while bands (Dentaform, Zahn 36, size 23/Roth 22, Dentaurum), brackets (equilibrium^®^ 2, φ 0.56 × 0, 76 mm/22 × 30, Roth 22, Dentaurum), and ligatures (remanium^®^, short, soft, φ 0.25 mm/10, Dentaurum) were made of stainless steel. Detailed alloy composition is given in our previous work [22].

### 4.2. Methods

#### 4.2.1. Preparation of the Test Samples

All tested samples including the control were prepared in artificial saliva, according to the Tani–Zucchi receipt pH 4.8 (composition: 1.5 g/L KCl; 1.5 g/L NaHCO_3_; 0.5 g/L NaH_2_PO_4_ × H_2_O; 0.5 g/L KSCN; 0.9 g/L lactic acid). This pH value (4.8) corresponds to the pH of one- and two-day-old dental biofilm and serves to simulate patients with poor oral hygiene [5]. A low pH associated with a high lactic acid content also tends to be associated with a high caries rate [6].

Eluates in artificial saliva were prepared according to the current ISO standard (ISO 10993-5:2009) [47], briefly, in this way: one orthodontic archwire, ten brackets, ten ligatures, and two bands were immersed individually in 20 mL of artificial saliva and autoclaved at 121 °C for 15 min (CertoClav, Leonding, Austria) and then incubated under sterile conditions on a rotary shaker (37 °C, 100/min, Unimax 1010, Heidolph, Schwabach, Germany) for three, seven, and fourteen days, respectively. For each time interval and each orthodontic appliance, five samples were prepared. For the purpose of this experiment, samples from the same time interval were combined in a new sterile vessel (100 mL total) to obtain a concentrated sample from which working concentrations were prepared. From each sample, 7 mL were taken to determine the released metal ions using an inductively coupled plasma mass spectrometer. Remainder of each sample (approximately 93 mL) was concentrated on a rotary evaporator (RV 10, IKA, Staufen, Germany). The resulting volume of each sample was concentrated 20-fold and used as stock solution to prepare the working solution (concentration range: 2.0×, 1.0×, 0.5×, 0.1×). All dilutions were prepared from a concentrated saliva stock solution using RPMI-1640 culture medium (Sigma-Aldrich-Chemie, Steinheim, Germany). Artificial saliva used as a control was prepared in the same way (concentrated 20× and diluted accordingly).

#### 4.2.2. Cytotoxic and Prooxidant Effects of Metals in Artificial Saliva Eluted from Dental Alloys

In brief, the cell lines CAL 27, AGS, CaCo2, and HepG2 were seeded in transparent multiwell microtiter plates (96 wells) for the cytotoxic assay and in black multiwell microtiter plates for the prooxidant assay. The concentration of cells in each well was 5 × 10^4^ cells/mL. After attachment, cells were treated with prior prepared eluates of parts of orthodontic appliances (archwires, brackets, bands, and ligatures) in four different concentrations of artificial saliva (0.1×, 0.5×, 1.0×, and 2×) for 24 h. Treatment with artificial saliva was used as a control. Each concentration was tested in six replicates, and each experiment was repeated three times.

The cytotoxic effect of the prepared samples was determined by the neutral red method [48]. After incubation and removal of eluates, cells were washed and 50 µM neutral red was added. The excess amount of neutral red was discarded, the cells were washed with PBS buffer, and the dye accumulated in the lysosomes was extracted with an extraction solution (ethanol:glacial acetic acid:water = 50:1:49, *v*/*v*/*v*). Absorbance was measured at 540 nm, and the percentage of surviving cells was calculated using the following formula:% of cell survival = (A_540_ nm sample/A_540_ nm control) × 100

The prooxidant effect of the artificial saliva samples was determined by the DCFH-DA method. After treatment of the cells, a 50 µM solution of DCFH-DA was added to the cells. After 30 min of incubation, fluorescence was measured. The fluorescence intensity was proportional to the free radicals formed in the cells [49].

Sample fluorescence was normalized to cell survival. Prooxidant effect was expressed as a percentage of sample fluorescence compared to control fluorescence:% prooxidant effect = (sample fluorescence/control fluorescence) × 100

#### 4.2.3. Effect of Dental Alloy Eluates on Hydroxyl Radical-Mediated DNA Strand Breakage

DNA damage was measured by the conversion of supercoiled φX-174 RF I double-stranded DNA into open circular and linear forms [50]. Briefly, the plasmid φX-174 RF I is in the form of supercoiled circular DNA and converts to a coiled form in the presence of the hydroxyl radical, which travels much more slowly through the gel than the supercoiled form. Hydrogen peroxide cannot induce the formation of hydroxyl radicals, but in the presence of metals, this form of this type of free radical can be formed and cause linearization of the plasmid [41,50,51]. Consequently, two bands of different intensity are formed in the gel after electrophoresis; the lower band represents the supercoiled DNA and the upper band represents the coiled (damaged) DNA. The ratio between the lower and upper bands and comparison with the negative control indicates the potential for the eluate from the dental alloy to induce DNA damage. The reaction mixtures contained 0.3 μg/mL of the plasmid, 0.3% hydrogen peroxide, and artificial saliva samples at a final concentration of 0.1×, 0.5×, 1.0×, and 2.0×. The reaction mixture was prepared in TE buffer. After incubation at 37 °C for 30 min, loading dye was added, and samples were immediately loaded into a 1% agarose gel containing 40 mM Tris, 20 mM sodium acetate, and 2 mM EDTA, and then electrophoresed in a horizontal slab-gel apparatus in Tris/acetate/EDTA gel buffer at 150 V for 60 min. After electrophoresis, the gels were stained with a 0.5 mg/mL solution of ethidium bromide for 2 h and then destained in water. The gels were then photographed under ultraviolet illumination and quantified using GelAnalyzer 19.1 software.

#### 4.2.4. Comet Assay

The comet assay was performed under alkaline conditions as described by Azqueta and Collins [52], with some modifications. Two replicate slides were prepared per sample. Agarose gels were prepared on fully frosted slides coated with 1% and 0.6% normal melting point (NMP) agarose. Cells were seeded in Petri dishes, Φ 5 cm, at a concentration of 5 × 10^4^ cells/mL and treated for 24 h with selected concentrations of dental alloy samples. After treatment, 10 μL of the cell suspension was mixed with 0.5% low melting point (LMP) agarose, placed on slides, and covered with a layer of 0.5% LMP agarose. The slides were immersed in freshly prepared ice-cold lysis solution (2.5 M NaCl, 100 mM Na_2_EDTA, 10 mM Tris—1% Na-sarcosinate, pH 10, all components provided by Sigma Aldrich, Oakville, Canada) containing 1% Triton X-100 (Sigma Aldrich, Oakville, Canada) and 10% dimethyl sulfoxide (Kemika, Zagreb, Croatia) for 1 h. Alkaline denaturation and electrophoresis were performed at 4 °C under dim light in freshly prepared electrophoresis buffer (300 mM NaOH, 1 mM Na_2_EDTA (Honeywell, Charlotte, NC, USA), pH 13.0). After 20 min of denaturation, the slides were randomly placed side by side in the horizontal gel electrophoresis tank, facing the anode. Electrophoresis at 25 V continued for an additional 20 min. After electrophoresis, the slides were gently washed three times at five-minute intervals with neutralization buffer (0.4 M Tris-HCl, pH 7.5). The slides were stained with ethidium bromide (20 μg/mL) and stored at 4 °C in humidified, sealed containers until analysis. Each slide was examined at 250× magnification using a fluorescence microscope (Zeiss, Oberkochen, Germany) equipped with a 515–560 nm excitation filter and a 590 nm blocking filter. The microscope image was transferred to a computerized image analysis system (Comet Assay II, Perceptive Instruments Ltd., Bury Saint Edmunds, UK). The comet parameters analyzed were tail length and tail DNA content. A total of 100 comets per sample (50 from each of the two replicate slides) were scored. Slides were blinded prior to analysis.

#### 4.2.5. Statistical Analysis

For the comparison of prooxidant effects and cytotoxicity, analysis of variance (ANOVA) was used together with the Student–Newman–Keuls post hoc test (using the commercial software IBM SPSS Statistics for Windows, Version 22.0 (IBM Corp., Armonk, NY, USA)). For comparison of genotoxicity, analysis of variance (ANOVA) was used together with the LSD test for the comet assay and the Mann–Whitney U test for the plasmid φX-174 RF I assay (using the Statistica data analysis software system, version 13.4.04; Tibco Software Inc., Palo Alto, CA, USA). PCA was used to determine the potential correlation between the orthodontic appliance and the analyses performed, using the Statistica software system (version 13.4.04; Tibco Software Inc., Palo Alto, CA, USA).

## 5. Conclusions

The effects of fixed orthodontic appliances on various cell lines are being studied intensively, and numerous clinical trials are being conducted, but each trial is different and sheds only a small light on the overall picture. Research in this area is still very thorough, useful, and intense, and, on the other hand, there is still much to discover. This type of research, using a fixed orthodontic appliance as would be used in the oral cavity and using eluates from such an appliance with known metal content and composition for toxicity research, brings us closer to more accurate in vivo conditions. In addition, this study uses cells from the tongue, stomach, liver, and colon, which are selected to mimic the passage of saliva through the gastrointestinal tract.

By deciding to conduct the study at a low pH (4.8), we focused on patients with poor oral hygiene, plaque accumulation, and altered acidity, which further compromised alloy stability. The average exposure time (for orthodontic treatments) is 18–24 months. Such long-term contact of alloys with the oral mucosa is not negligible, and it is possible that different cellular reactions may develop under real conditions. Under certain circumstances, there may be an accumulation of ions due to other intakes (food, beverages, piercing, etc.), which may lead to a significant synergistic effect and, consequently, to intense effects on human health. For this reason, the information on cytotoxicity should not be disregarded.

## Figures and Tables

**Figure 1 ijms-24-09940-f001:**
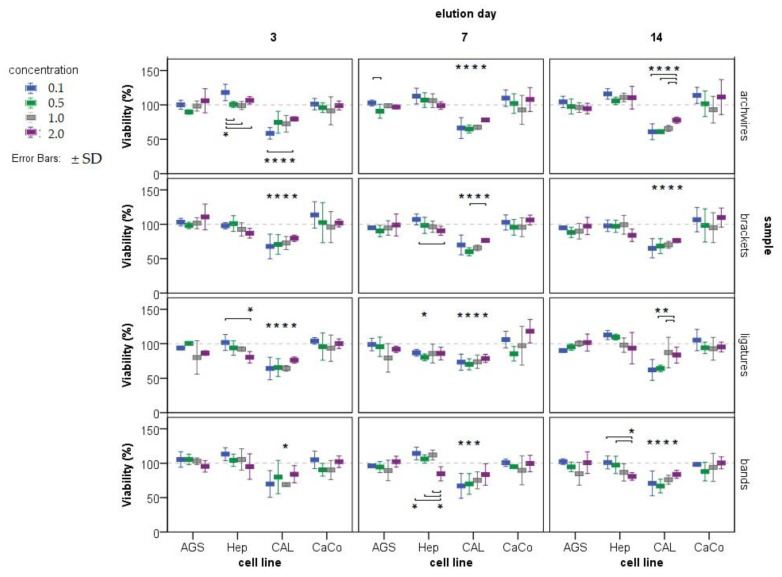
Estimation of cytotoxicity (expressed as cell viability in %) in four cell lines (AGS, Hep-G2, CAL 27, and CaCo-2) after exposure to eluates of metal ions released from various parts of fixed orthodontic appliances (archwires, brackets, ligatures, and bands) at three different immersion periods (3, 7, and 14 days) and four different concentrations of each eluate (0.1×; 0.5×, 1.0×, and 2.0×, represented by different colored boxes). Results are given as mean ± SD. The control group (treated with artificial saliva only) is set at 100% and presented as a dashed line. * indicates a concentration at which viability (%) is significantly different from 100% (control sample), each * is located directly above the data to which it refers, and refers only to that information and not to others in that group; more * means that more concentrations tested are significantly different from 100%; lines/connections are associated concentrations at which cell viability differs significantly (Student–Newman–Keuls post hoc test, *p* < 0.05).

**Figure 2 ijms-24-09940-f002:**
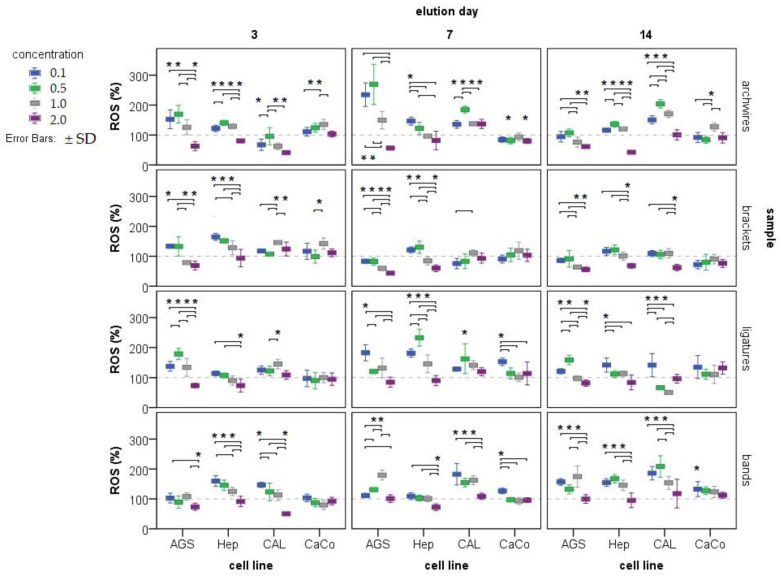
The presence of free radicals was tested in four cell lines (AGS, Hep-G2, CAL 27, and CaCo-2) after exposure to eluates of metal ions released from various orthodontic appliances (archwires, brackets, ligatures, and bands) at three different immersion periods (3, 7, and 14 days) and four different concentrations of each eluate (0.1×; 0.5×, 1.0×, and 2.0× represented by different colored boxes). Results are expressed as a percentage of sample fluorescence relative to the fluorescence of the control (artificial saliva). The control group (treated with artificial saliva only) is set at 100% and presented as a dashed line. Results are given as mean ± SD. * indicates a concentration at which ROS content is significantly different from 100%; each * is located just above the data to which it refers, and refers only to that information and not to others in that group; more * means that more concentrations tested are significantly different from 100%; lines/connections are associated concentrations at which produced ROS differ significantly (Student–Newman–Keuls post hoc test, *p* < 0.05).

**Figure 3 ijms-24-09940-f003:**
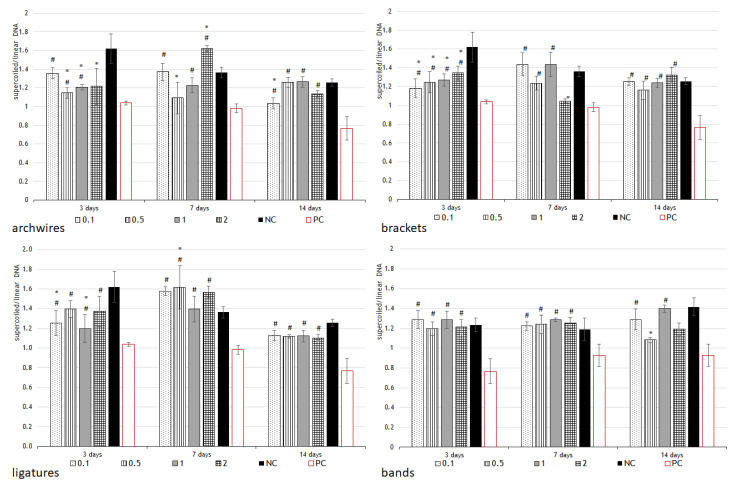
Testing for DNA damage. Eluates prepared from various parts of fixed orthodontic appliance (archwires, brackets, ligatures, and bands) after three immersion periods (3, 7, and 14 days) at four different concentrations (0.1×; 0.5×, 1.0×, and 2.0×) were used to test DNA stability. The plasmid φX-174 RF I was used for this experiment. The plasmid in the form of supercoiled circular DNA converts to a linear form in the presence of metal ions and H_2_O_2_, producing hydroxyl radicals. After electrophoresis, two bands of different intensities are formed in the gel; the upper band represents the coiled (damaged, converted to linear form) DNA, and the lower band represents the supercoiled DNA. The ratio between the lower and upper bands and comparison to the negative control indicates the potential for the dental alloy eluate to cause DNA damage. Negative control (NC)—plasmid + hydrogen peroxide; positive control (PC)—plasmid + hydrogen peroxide + UV irradiation for 15 min. Results are expressed as mean ± SD. * indicates significantly different compared to the negative control; # indicates significantly different compared to the positive control. Mann–Whitney U test, *p* < 0.05.

**Figure 4 ijms-24-09940-f004:**
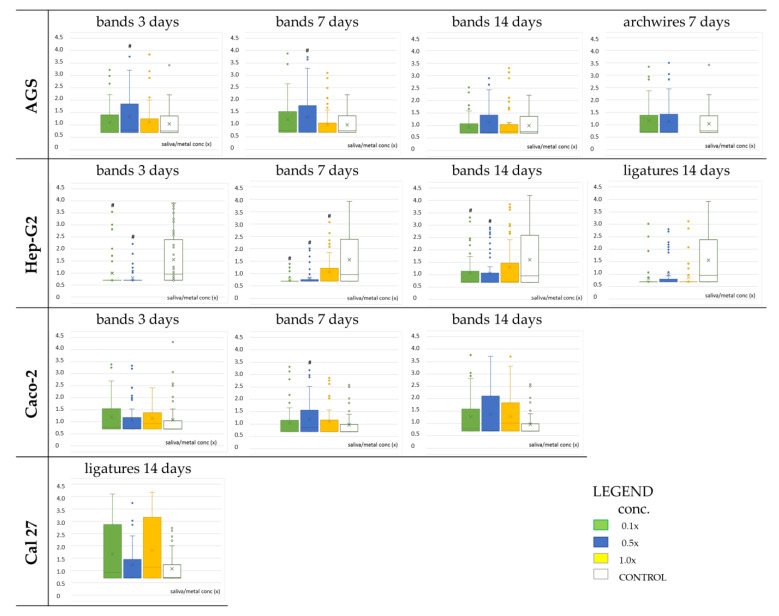
Genotoxicity (comet assay: tail moment) in cultures of selected cell lines induced by eluates obtained from bands immersed for 3, 7, and 14 days, ligatures 14 days, and archwires 7 days, and three different concentrations of each eluate (0.1×; 0.5×, and 1.0× represented by different colored boxes) # indicates significantly different compared to the control (LSD test, *p* < 0.05).

**Figure 5 ijms-24-09940-f005:**
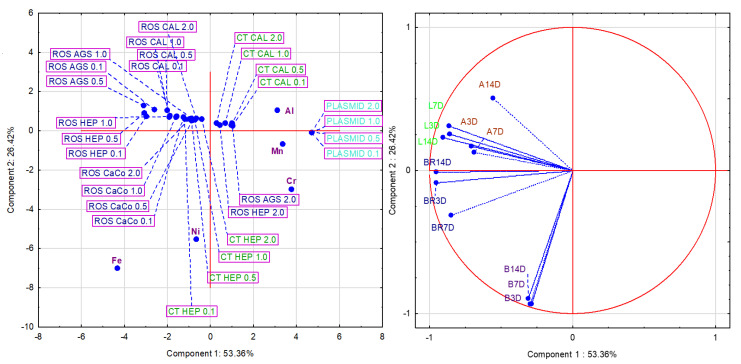
Principal component analysis (PCA) applied to orthodontic appliances (archwires (A), bands (B), ligatures (L), and brackets (BR) for 3, 7, and 14 days), five released metals (Fe, Cr, Ni, Mn, and Al), cytotoxicity (CT) in two different cell lines (CAL 27 and Hep 2 cells); and four different dilutions (0.1, 0.5, 1.0, and 2.0), the presence of free radicals (ROS) in all four cell lines, and all tested dilutions (0.1, 0.5, 1.0, and 2.0) and DNA damage (plasmid) in four dilutions (0.1, 0.5, 1.0, and 2.0) in the experimental group. The planes correlate components 1 and 2 (with eigenvalue 6.40 vs. 3.17). The left side of the figure projects the cases, while the right side shows the variables.

**Table 1 ijms-24-09940-t001:** Eigenvalues of the principal component analysis (PCA) with total variance analysis for orthodontic appliances (archwires, bands, ligatures, and brackets, for 3, 7, and 14 days), five released metals (Fe, Cr, Ni, Mn, and Al), cytotoxicity in four different cell lines (CAL cells—dilutions 0.1×, 0.5×, 1.0×, and 2.0×; Hep 2 cells—dilutions 1.0× and 2.0×; AGS cells—dilutions 1.0× and 2.0×; and CaCo2 cells—1.0×), the presence of free radicals in four cell lines (CAL cells—dilutions 1.0× and 2.0×; Hep 2 cells—dilutions 1.0× and 2.0×; AGS cells—dilutions 1.0× and 2.0×; and CaCo2 cells—dilutions: 0.1×, 1.0×, and 2.0×), and DNA damage in four dilutions (0.1×, 0.5×, 1.0×, and 2.0×) in the experimental group. The number of components is five.

Component	Eigenvalue	Total Variance (%)	Cumulative Eigenvalue	Cumulative (%)
1	6.40	53.36	6.40	53.36
2	3.17	26.42	9.57	29.78
3	1.17	9.57	10.74	89.57
4	0.54	4.54	11.29	94.08
5	0.24	2.04	11.53	96.12

**Table 2 ijms-24-09940-t002:** Eigenvector of correlation matrix for orthodontic appliances (archwires, brackets, ligatures, and bands for 3, 7, and 14 days), five released metals (Fe, Cr, Ni, Mn, and Al), cytotoxicity in four different cell lines (CAL cells—dilutions 0.1×, 0.5×, 1.0×, and 2.0×; Hep 2 cells—dilutions 1.0× and 2.0×; AGS cells—dilutions 1.0× and 2.0×; and CaCo2 cells—1.0×), the presence of free radicals in four cell lines (CAL cells—dilutions 1.0× and 2.0×; Hep 2 cells—dilutions 1.0× and 2.0×; AGS cells—dilutions 1.0× and 2.0×; and CaCo2 cells—dilutions 0.1×, 1.0×, and 2.0×), and DNA damage in four dilutions (0.1×, 0.5×, 1.0×, and 2.0×) in the experimental group.

Variable	Component 1	Component 2	Component 3
**archwires 3 days**	−0.71	0.17	0.59
**archwires 7 days**	−0.69	0.13	0.67
**archwires 14 days**	−0.56	0.51	0.09
**brackets 3 days**	−0.95	−0.09	−0.23
**brackets 7 days**	−0.85	−0.31	−0.36
**brackets 14 days**	−0.95	−0.01	−0.15
**ligature 3 days**	−0.91	0.23	−0.10
**ligature 7 days**	−0.87	0.31	−0.12
**ligature 14 days**	−0.86	0.26	−0.17
**bands 3 days**	−0.29	−0.93	0.12
**bands 7 days**	−0.30	−0.94	−0.11
**bands 14 days**	−0.31	−0.90	0.28

## Data Availability

The data presented in this study are available on request from the corresponding author.

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
