# Peer review of "Toxicity of Metal Ions Released from a Fixed Orthodontic Appliance to Gastrointestinal Tract Cell Lines"

_ijms, 2023, doi:10.3390/ijms24129940_

Round 1

Reviewer 1 Report (Previous Reviewer 3)

Dear authors, thank you for your efforts to improve the manuscript. 

Author Response

Thank you very much.

Reviewer 2 Report (New Reviewer)

The authors measured the response of four different cancer cells (tongue, liver, stomach, and colon) to metal ions, which released by fixed orthodontic appliances (eluates of brackets, archwires, ligatures, and bands), using various assays. However, there still are concerns that need to be addressed. 

1. In Figure 1, the figure legend mentioned that “* indicates a concentration at which ROS content is significantly different from 100% (shown with a dashed line); each * is located just above the data to which it refers, and refers only to that information and not to others in that groups.” However, a). this figure is related to cytotoxicity, not for ROS. b). it is not clear what “significantly different from 100%” mean. Did author normalize to empty control (no treatment)? c). The labeling is also not clear and confused. 

2. Figure 2 depicts the same issue as Figure 1. It is not clear what “significantly different from 100%” mean. What was the comparison? compared to no treatment control or anything else?

3. The ROS content are not consistent to cell viability, that higher ROS content indicates less cell viability. What is the interpretation from authors?

4. In Figure 3, the image quality of gel should be improved. In the bar chart, the author described 6 groups, however, only 4 groups are presented in the gel image. What the number on y axis indicates? Relative fold change to control? or the ratio of linear to circle? 

5. In Figure 4, the organization of panel should be improved to help audience easily understand. 

Author Response

  1. In Figure 1, the figure legend mentioned that “* indicates a concentration at which ROS content is significantly different from 100% (shown with a dashed line); each * is located just above the data to which it refers, and refers only to that information and not to others in that groups.” However, a). this figure is related to cytotoxicity, not for ROS. b). it is not clear what “significantly different from 100%” mean. Did author normalize to empty control (no treatment)? c). The labeling is also not clear and confused. 

Thank you very much for this comment. You are absolutely right.

  1. The authors made a mistake – we corrected that.
  2. Indeed, the meaning was not clear. Everything is normalized to the control (samples treated with artificial saliva only). That’s why this sentence is added above the figure 1.

“The control group (treated with artificial saliva only) is set at 100%, and present with a dashed line. * indicates a concentration at which viability (%) is significantly different from 100% (control sample).”

3. 4 different colors for every dilution (concentration tested: 0.1x, 0.5x, 1.0x and 2.0x) were used. 4 different cell lines – presented in vertical lines, for every elution period (3, 7 or 14 days) presented in small separate figure. Authors are not sure what is confusing so, we do not how to further improve figure.

  1. Figure 2 depicts the same issue as Figure 1. It is not clear what “significantly different from 100%” mean. What was the comparison? compared to no treatment control or anything else?

Thank you for the comment. The text is improved in the same way as for figure 1.

  1. The ROS content are not consistent to cell viability, that higher ROS content indicates less cell viability. What is the interpretation from authors?

Reactive oxygen species, can control the fundamental process of cell division, which, can lead to uncontrolled cell growth and cancer. On the other hand, the high level of free radicals can trigger cell death resulting in the decrease of the cell viability. The main finding that goes in this direction is the paper published by J. Mansfeld who proved that ROS can induce a key protein for cell division, CDK2. They also find out that CDK2 can be oxidised and that this oxidation can contribute to development a new class of CDK2-specific inhibitors that do not target closely related kinases. Obviously, the concentration of free radicals play significant role in promotion of cell division or induction of cell death respectively. In this paper, as was shown on tongue cells, that were the most sensitive cells and all saliva samples tested were toxic to this cell line at all concentration ranges tested, a most intriguing and interesting response of CAL27 cells to metal ion treatment was noted. For most samples, viability was lowest at the lowest dose tested (0.1x), with a tendency to increase at a dose of 2x. The results demonstrate involvement of different mechanisms while cells cope with smaller or greater quantity of metal ions. This is additionally proved with analysis that made a part of new publications/papers.

  1. In Figure 3, the image quality of gel should be improved. In the bar chart, the author described 6 groups, however, only 4 groups are presented in the gel image. What the number on y axis indicates? Relative fold change to control? or the ratio of linear to circle? 

Thank you for the comment. The quality of the gels can not be improved. Besides, these figures are not so important – the authors use them as a prove/or only to emphasize the difference between the samples.

So, the authors decide to move those gel figures from the charts. 

Name for y ax is added to the charts.

  1. In Figure 4, the organization of panel should be improved to help audience easily understand. 

Figure 4 is reorganized. Hopefully, to more understandable way.

In order to check, improve and additionally clarify the text, manuscript is read again and minor corrections were done (please check Track-changes).

Round 2

Reviewer 2 Report (New Reviewer)

Accept in present form.

This manuscript is a resubmission of an earlier submission. The following is a list of the peer review reports and author responses from that submission.

Round 1

Reviewer 1 Report

The manuscript needs minor revision.

Pleases add Materials & Methods section before the Results section.

 Discussion: Pleases add the limitations of the study at the end of the discussion section.

Conclusions: Please summarize the conclusion section.

References: Most of the references are old. Please increase the number of more recently studies in the introduction and discussion sections

Author Response

The authors thank the reviewer for his comments and suggestions.

First, we must apologize for the error that occurred when uploading the document - there was no Figure 1, only the accompanying text.

In revising the references, we need to add some new sentences that refer to newly added references.

To clarify and improve the manuscript, we have also added some sentences for additional explanation of the data presented in Table 2.

All these sentences are marked in red.

Pleases add Materials & Methods section before the Results section.

According to the instructions of the Journal, the Results section precedes the Materials and Methods section.

 Discussion: Pleases add the limitations of the study at the end of the discussion section.

This is added to the Discussion part:

The only limitation of the study that the authors were able to identify is the fact that the experiments were performed on cell lines that were different from normal cells. However, a negative control was included in all experiments, and all results obtained are presented in comparison with the control. All observed changes are the result of exposure to the eluates and the different metal concentration in the eluates. Finally, the use of cell lines in toxicological studies is widely accepted as a biological test system, and the results can be well interpreted by appropriate statistical analysis.

Conclusions: Please summarize the conclusion section.

This is added to the Conclusion part:

The effects of fixed devices on cells of the oral mucosa and gastrointestinal tract are being intensively researched, and numerous clinical trials are being conducted, but each trial is different and sheds only a small light on the overall picture. Research in this area is still very thorough and intense, and much remains to be discovered.

In our study, we tried to get as close as possible to in vivo conditions. We focused on the patient group where the highest degree of corrosion was expected due to poor oral hygiene and altered acidity of the medium, and adjusted the pH to these conditions.

We believe that the information about cytotoxicity should not be ignored, because in certain circumstances there is an accumulation of ions due to other intakes (food, beverages, piercing, etc.) and a significant synergistic effect may occur.

References: Most of the references are old. Please increase the number of more recently studies in the introduction and discussion sections

We thank you for this comment. You are right, the references were old. The authors reviewed all the references in detail to see what could be replaced. We tried to replace all the references we could find, which was often difficult because the original citation referred to the essentials and each newer reference was much more detailed and could not fully replace the outcome one. So we replaced 9 references with newer ones and removed several not so important references. Now there are 30 references from 2013 to present (out of 43 total).

Here are new 9 references:

  1. Proffit, W. R., Fields, H. W., Sarver, D. M. Contemporary orthodontics. 6th Philadelphia: Elsevier; 2018, 599-603. Contemporary Orthodontics (elsevier.com)
  2. Ferrer, MD, Pérez, S., Lopez, AL., Sanz, JL., Melo, M., Llena, C., Mira, A. Evaluation of Clinical, Biochemical and Microbiological Markers Related to Dental Caries. J. Environ. Res. Public Health 2021, 18, 6049. https://doi.org/10.3390/ijerph18116049
  3. Anjos, V. A., da Silva-Júnior F. M.R., Souza, M. Cell damage induced by copper: An explant model to study anemone cells. Toxicol in vitro. 2017, 28 (3), 365-372. https://doi.org/10.1016/j.tiv.2013.11.013
  4. Castro, S. M., Ponces, M. J., Lopes, J. D., Vasconcelos, M., Pollmann, M. C.F., Orthodontic wires and its corrosion—The specific case of stainless steel and beta-titanium J Dental Sci 2015, 10(1), 1–7. https://doi.org/10.1016/j.jds.2014.07.002
  5. Hsu, M. Y., Mina, E., Roetto, A., Porporato, P. E., Iron: An Essential Element of Cancer Metabolism. Cells, 2020, 9(12), 2591. https://doi.org/10.3390/cells9122591.
  6. Angelé-Martínez, C., Goodman, C., Brumaghim, J. Metal-mediated DNA damage and cell death: mechanisms, detection methods, and cellular consequences. Metallomics, 2014. 6(8), 1358–1381. https://doi.org/10.1039/c4mt00057a
  7. Song, H-S., Sung, S-H., Jin, Y-W., Kim, C-S., Kim, Y-S., Bang, W-G., Chung, N-H. Mobilization of Iron into Cell from Ambient Particulate Matter and Its Possible Participations to DNA Single Strand Break. J Appl. Biol. Chem. 2004. 47(2), 65-70.
  8. Green, M. R., Sambrook, J. Analysis of DNA by Agarose Gel Electrophoresis. Cold Spring Harb. Protoc. 2019, (1), pdb.top100388. https://doi.org/10.1101/pdb.top100388.
  9. Azqueta A, Collins AR. The essential comet assay: a comprehensive guide to measuring DNA damage and repair. Arch Toxicol. 2013, 87(6), 949-968. https://doi.org/10.1007/s00204-013-1070-0.

Reviewer 2 Report

Dear Editors :

The novelty of the study is very low because lots of clinical trials have already shown no genotoxic effects of fixed appliances on oral mucosal cells and gastrointestinal cells.   Therefore, I suggested rejecting the manuscript. 

Author Response

The authors thank the reviewer for his comments and suggestions.

First, we must apologize for the error that occurred when uploading the document - there was no Figure 1, only the accompanying text.

The novelty of the study is very low because lots of clinical trials have already shown no genotoxic effects of fixed appliances on oral mucosal cells and gastrointestinal cells.   Therefore, I suggested rejecting the manuscript. 

You are right that many things are being researched and that many clinical studies are being conducted on the effects of fixed appliances on oral mucosal cells and gastrointestinal cells, but each experiment is actually different and only illuminates a small part of the whole picture. Research in this area is still very thorough and intense, and there are more things that should be discovered.

We have studied not only the genotoxic, but also the cytotoxic and prooxidant damage of various metal ions eluted from orthodontic devices during different elution periods in four different cell lines (carefully selected to answer the question of how released metal ions affect the epithelium of the oral cavity and stomach, which are the first to come into contact with the eluted metals, as well as hepatocytes and the intestine). We have shown how the release of metal ions occurs - unpredictably and even with metals that should not be found because they are not part of a particular device, and the amount of all metals released. Many previous studies have been done either with wires only or with brackets, in different media, and with different protocols. In our study, we tried to get as close as possible to in vivo conditions. We focused on the group of patients in whom the highest degree of corrosion was expected due to poor oral hygiene and altered acidity of the media, and adjusted the pH to these conditions.

We believe that information about cytotoxicity should not be ignored, because in certain circumstances there is an accumulation of ions due to other intakes (food, beverages...) and a synergistic effect may occur with enhanced effect...

Moreover, there is a possibility that even non-toxic concentrations of cations released from dental alloys may be sufficient to cause biological changes.

Reviewer 3 Report

In the current article, the authors investigate the unpredictable release of heavy metals from fixed orthodontic appliances (dental alloys and bands) and how the elution process influences the cell lines: CAL27, Hep-G2,  Caco-2, and AGS. The composition of the eluates is invastigated for ROS, prooxidative DNA damage, cytotoxicity, and genotoxicity of metal ions eluted on the type and amount of metal ions released by orthodontic appliances during different periods of elution.

The article is well-structured and it is prepared according to the instructions of the journal. The subject is clearly identified and the manuscript seems informative and theoretical. The discussion and conclusion are logically based and directly related to the presented results.

A large part of the cited articles in the references section and respectively the main text are from the old period (before 2013's years). I suggest that authors choose more recent articles, with the percentage of new articles being approximately 70% of total citations.

Author Response

The authors thank the reviewer for his comments and suggestions.

First, we must apologize for the error that occurred when uploading the document - there was no Figure 1, only the accompanying text.

In revising the references, we need to add some new sentences that refer to newly added references.

To clarify and improve the manuscript, we have also added some sentences for additional explanation of the data presented in Table 2.

All these sentences are marked in red.

In the current article, the authors investigate the unpredictable release of heavy metals from fixed orthodontic appliances (dental alloys and bands) and how the elution process influences the cell lines: CAL27, Hep-G2,  Caco-2, and AGS. The composition of the eluates is invastigated for ROS, prooxidative DNA damage, cytotoxicity, and genotoxicity of metal ions eluted on the type and amount of metal ions released by orthodontic appliances during different periods of elution.

The article is well-structured and it is prepared according to the instructions of the journal. The subject is clearly identified and the manuscript seems informative and theoretical. The discussion and conclusion are logically based and directly related to the presented results.

A large part of the cited articles in the references section and respectively the main text are from the old period (before 2013's years). I suggest that authors choose more recent articles, with the percentage of new articles being approximately 70% of total citations.

We thank you for this comment. You are right, the references were old. The authors reviewed all the references in detail to see what could be replaced. We tried to replace all the references we could find, which was often difficult because the original citation referred to the essentials and each newer reference was much more detailed and could not fully replace the outcome one. So we replaced 9 references with newer ones and removed several not so important references. Now there are 30 references from 2013 to present (out of 43 total).

Here are new 9 references:

  1. Proffit, W. R., Fields, H. W., Sarver, D. M. Contemporary orthodontics. 6th Philadelphia: Elsevier; 2018, 599-603. Contemporary Orthodontics (elsevier.com)
  2. Ferrer, MD, Pérez, S., Lopez, AL., Sanz, JL., Melo, M., Llena, C., Mira, A. Evaluation of Clinical, Biochemical and Microbiological Markers Related to Dental Caries. J. Environ. Res. Public Health 2021, 18, 6049. https://doi.org/10.3390/ijerph18116049
  3. Anjos, V. A., da Silva-Júnior F. M.R., Souza, M. Cell damage induced by copper: An explant model to study anemone cells. Toxicol in vitro. 2017, 28 (3), 365-372. https://doi.org/10.1016/j.tiv.2013.11.013
  4. Castro, S. M., Ponces, M. J., Lopes, J. D., Vasconcelos, M., Pollmann, M. C.F., Orthodontic wires and its corrosion—The specific case of stainless steel and beta-titanium J Dental Sci 2015, 10(1), 1–7. https://doi.org/10.1016/j.jds.2014.07.002
  5. Hsu, M. Y., Mina, E., Roetto, A., Porporato, P. E., Iron: An Essential Element of Cancer Metabolism. Cells, 2020, 9(12), 2591. https://doi.org/10.3390/cells9122591.
  6. Angelé-Martínez, C., Goodman, C., Brumaghim, J. Metal-mediated DNA damage and cell death: mechanisms, detection methods, and cellular consequences. Metallomics, 2014. 6(8), 1358–1381. https://doi.org/10.1039/c4mt00057a
  7. Song, H-S., Sung, S-H., Jin, Y-W., Kim, C-S., Kim, Y-S., Bang, W-G., Chung, N-H. Mobilization of Iron into Cell from Ambient Particulate Matter and Its Possible Participations to DNA Single Strand Break. J Appl. Biol. Chem. 2004. 47(2), 65-70.
  8. Green, M. R., Sambrook, J. Analysis of DNA by Agarose Gel Electrophoresis. Cold Spring Harb. Protoc. 2019, (1), pdb.top100388. https://doi.org/10.1101/pdb.top100388.
  9. Azqueta A, Collins AR. The essential comet assay: a comprehensive guide to measuring DNA damage and repair. Arch Toxicol. 2013, 87(6), 949-968. https://doi.org/10.1007/s00204-013-1070-0.

Round 2

Reviewer 2 Report

I still insisted on my previously suggestions.